# Design and Control of an Underwater Robot Based on Hybrid Propulsion of Quadrotor and Bionic Undulating Fin

Xiaofeng Zeng [1], Minghai Xia [1], Zirong Luo [1,*], Jianzhong Shang [1,*], Yuze Xu [1] and Qian Yin [2]

1 College of Intelligence Science and Technology, National University of Defense Technology, Changsha 410073, China
2 College of Energy and Power Engineering, Changsha University of Science & Technology, Changsha 410076, China
* Correspondence: luozirong@nudt.edu.cn (Z.L.); shangjianzhong@nudt.edu.cn (J.S.)

**Abstract:** Stable, quiet, and efficient propulsion methods are essential for underwater robots to complete their tasks in a complex marine environment. However, with a single propulsion method, such as propeller propulsion and bionic propulsion, it is difficult to achieve high efficiency and high mobility at the same time. Based on the advantages of the high-efficiency propulsion of a bionic undulating fin and the stable control of the propeller, an underwater robot based on the hybrid propulsion of a quadrotor and undulating fin is proposed in this paper. This paper first introduces the mechanical implementation of the underwater robot. Then, based on kinematic modeling and theoretical derivation, the underwater motion and attitude of the robot are analyzed and the 6-DOF dynamic equation of the robot is established. Finally, the underwater motion performance of the robot is verified through field experiments. The experimental results show that the robot can realize the heave motion, surge motion, and in-situ steering motion independently and can hover stably. When the undulating frequency is 6 Hz, the maximum propulsion speed of the robot can reach up to 1.2 m/s (1.5 BL/s).

**Keywords:** underwater robot; undulating fin; quadrotor; prototype experiment

## 1. Introduction

Underwater robots can replace or accompany human beings to complete underwater missions in unknown and complex marine environments efficiently, which have broad application prospects for scientific exploration, economic operations, and military fields.

Propulsion mode is the key factor to determine the underwater maneuverability, endurance, and concealment of the robot. Rotor and bionic propulsion are two mainstream propulsion modes.

Multi-rotor propulsion has the advantages of high maneuverability and stability, which is particularly important for underwater robots to perform submarine salvage, sampling, pipeline maintenance, and other operations. Pierrot discussed the distribution configuration of six rotors in several different underwater robots and obtained better distribution [1]. Choi developed an eight-rotor spherical underwater robot called Odin, which has good maneuverability and hydrodynamic performance [2]. These multi-thruster underwater vehicles often have six to ten thrusters, which are relatively redundant for basic motion. Drews proposed a hybrid unmanned aerial underwater vehicle and established its kinematic and dynamic model [3], which was the first time quadrotors were used in underwater environments. Ranganathan designed a quad-rotor underwater vehicle and analyzed it by establishing its mathematical model based on the Newton–Euler method [4]. In addition, quadrotor underwater vehicles can achieve flexible and stable motion with fewer thrusters. Bian designed an X-shaped quad-rotor underwater vehicle, modeled it, and analyzed its motion mode to show the benefits of setting the four thrusters as an X shape [5].

Bionic design is an effective method to develop underwater robots [6,7]. Bionic propulsion is mainly divided into two types: the body/caudal fin (BCF) and median/paired fin (MPF) propulsions, which have the advantages of low noise and efficiency. As shown in Figure 1a, the fish of the BCF mode bend their body into a backward-moving propulsive wave that extends to their caudal fin for producing the force of their forward propulsion. Cai developed a robot fish called Robo ray II with flexible swinging pectoral fins [8]. They found that the thrust coefficient increases with the Strouhal number and reached 0.56 at maximum. The maximal linear swimming speed of Robo ray II is about 0.5 times that of body length per second (BL/s). Costa et al. [9] designed a *Carangiform* swimming robot and found that the average cruising speed has a linear dependency on the motor rotation frequency and the invariance of the Strouhal number with respect to the motor frequency. Samuel et al. [10] investigated the *Carangiform* fish body and proposed multi-segment fins for describing the kinematics and performance optimization. They suggested that five-segment models can approximate the undulatory movements of (sub-) *Carangiform* swimmers during steady swimming, with at least 99% accuracy (model error < 0.01 L). Salazar et al. [11] analyzed the piezoelectric energy of the aquatic unmanned vehicles under the *Thunniform* condition of the fish's motion. The simultaneous inclusion of geometric and piezoelectric nonlinearities in the analysis offers a unique solution of useable power scavenged from the *Anguilliform*, *Subcarangiform*, *Carangiform*, and *Thunniform* motions. Zhang et al. [12] Samuel et al. invented a two-segment fin which imitates the *Ostraciidae* fish for swimming underwater. The resonant frequency and the thrust force increased about three times and were 12% higher when compared with the one-segment fin.

Unlike the BCF fish swimming mode, the undulating fins of the MPF fish keep their body rigid and use their median and pectoral fins which can generate propulsion forces in different directions as shown in Figure 1b. Although the propulsion velocity is lower compared with some BCF fish, the high efficiency and great maneuverability have made the MPF fish a good bionic sample for the design of the underwater propulsion devices. Zhou et al. [13] established the coupled CFD model of the underwater robots propelled by bionic undulating fins and compared the simulation and experimental results. Their work has formed a meaningful basis and computational platform for future studies on the propulsion mechanism and control algorithm of bionic underwater robots. Rahman verified the braking ability of the double undulating fin robot by measuring the stopping time and swimming distance of the robot after the braking frequency was applied to linear and rotary motion [14]. They confirmed that the undulating fin propulsion system can effectively perform braking even in complex underwater explorations. Wang et al. [15] designed and modeled a biomimetic stingray-like robotic fish. After a year, he [16] made a prototype and carried out underwater experiments to study the swimming performance of the robotic fish. The maximum velocity of the robotic fish is 4.3 cm/s (nearly 0.18 BL/s) at an oscillation frequency of 0.5 Hz. Scaradozzi et al. [17] proposed a partially biomimetic underwater robot with a hybrid propulsion system and developed it to overcome the challenges involved in improving the thrust efficiency.

Especially, *Gymnarchus niloticus*, as a kind of MPF fish, keep their body rigid and use their long fin to swim. Shahin et al. [18] proposed numerical and experimental methods to investigate their counter-propagating propulsion in order to improve maneuverability and stability. Liu and Izaak et al. [19,20] measured 3D flow structure and fields during the forward and station-keeping of undulating swimming through particle image velocimetry (PIV). Zhang et al. [21] proposed a modular mechanism with ten servomotors in order to drive the fin to undulate. They calculated the pressure distribution, the thrust force, and the efficiency of the undulating fins by numerical simulation and experimental measurements. Zhao et al. [22] investigated the hydrodynamic performance of the undulating ribbon fin combined with the sinusoidal swing and the fish body's undulating motion in a three-dimensional simulation. The effect of the phase-angle difference and angular amplitude were discussed to illustrate the thrust augmentation in the undulating fins with combined undulating motion. Low [23,24] developed the biomimetic robot NKF-II with

two mechanisms of layouts with cranks and slider links. The robot could generate arbitrary undulating waveforms and kinematic analyses were also performed. Hu and his colleagues studied the locomotion mechanism of bio-inspired robotic undulating fins by carrying out biological measurements [25], establishing kinematic equations, computational fluid dynamics (CFD) analyses [26], trajectory tracking of the fin's motion [27], and experimental analyses of the robot propelled by undulating fins. These works showed that the undulating fins based on the *Gymnarchus niloticus* could move forward, backward, or turn around underwater with good propulsive efficiency and high maneuverability.

In the field of marine search and rescue and marine exploration, it is hoped that marine robots will have the ability to move quietly and efficiently like fish, and have the ability to move in all directions like rotor robots to complete complex tasks and avoid disturbing other marine creatures, so as not to damage the environment.

To achieve this goal, we designed a robot based on hybrid propulsion of quadrotor and bionic undulating fins. The robot mainly works in the undulating fin-propulsion mode to achieve quiet and efficient movement and when the robot needs to hover, such as when performing underwater pipe maintenance tasks, the robot will work in the rotor mode to achieve static hover. In this paper, the mechanical structure of the robot is described in Section 2. The dynamic and control model of the robot is established and the motion control strategy is discussed in Section 3. The simulation and experiments are conducted in Section 4, which verify that the robot can realize five degrees of freedom motion, such as surge, heave, roll, pitch, and yaw. The experimental results show that the robot has good maneuverability and stability. When the undulating frequency is 6 Hz, the maximum propulsion speed of the robot can reach up to 1.2 m/s (1.5 BL/s). Finally, a conclusion is drawn in Section 5.

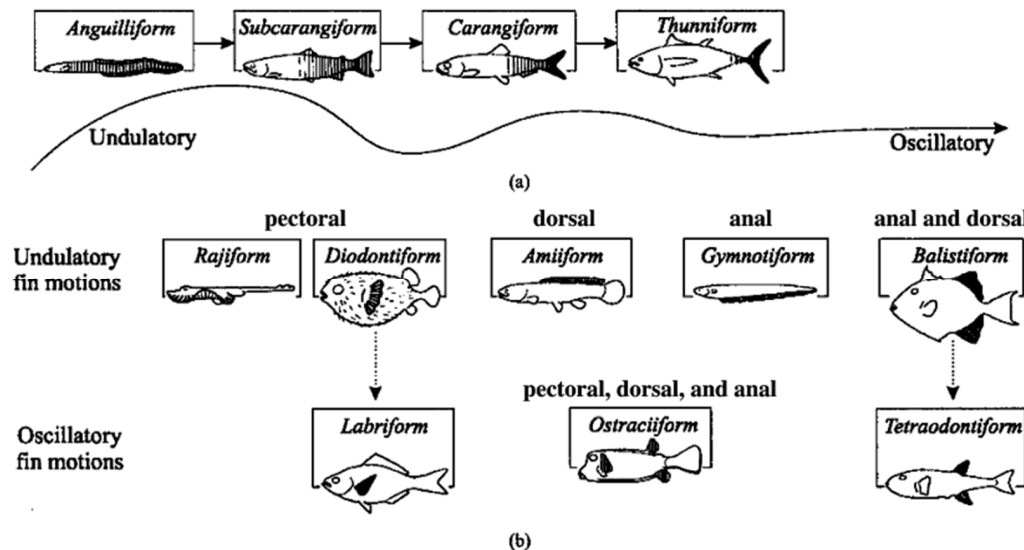

**Figure 1.** Swimming modes associated with (**a**) BCF propulsion and (**b**) MPF propulsion. Shaded areas contribute to thrust generation [28]. Reproduced with permission from J.E. Colgate, IEEE Journal of Oceanic Engineering; published by IEEE, 2004.

## 2. Mechanical Structure

Our goal is to design an operating robot that can move quietly and efficiently to its destination like a fish and then hover like a quadrotor robot in order to perform a variety of tasks such as maritime search and rescue, hazard disposal, pipeline maintenance, etc. The design model of the underwater robot is shown in Figure 2a. It is composed of one control cabin, four propellers, one undulating fin, one driving unit, one tilting unit, and one sine wave generator. The four propellers are configured as an X-type and installed vertically at the diagonal of the control cabin. The undulating fin is placed under the control cabin

linking with the sine wave generator. As shown in Figure 2b,c, the undulating fin can tilt left and right around the central axis to generate horizontal propulsion and yaw moment.

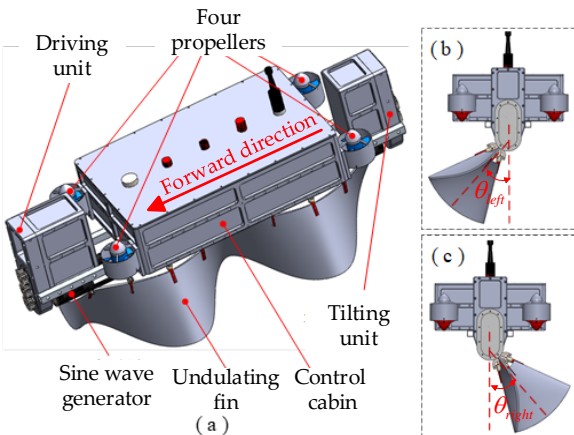

**Figure 2.** The overall design of the underwater robot. (**a**) Design model of the underwater robot; (**b**) Undulating fin tilts left side, $\theta_{left}$ is tilting angel; (**c**) Undulating fin tilts right side, $\theta_{right}$ is tilting angel.

At present, the undulating fins generally use a set of servo motors as the driving components [29], and each fin bar is driven by an electric motor. In this case, by coordinating and controlling the phase difference of each motor, the desired waveform can be formed. This kind of design is popular because of its simple transmission and multiple controllable parameters. However, too many motors bring some drawbacks, including the complexity of the mechanical structure, the increase in the difficulty of sealing, and the sacrifice of weight. Furthermore, system reliability will decrease significantly if one single motor fails which causes the failure of the entire waveform. Meanwhile, due to the limited response speed of the steering gear, the frequency of the undulating fin cannot reach a high level, and, generally, the maximum frequency is only 2–3 Hz [30].

In order to realize the high-frequency-driven undulating fin to study the driving performance of the undulating fin under high-frequency characteristics and improve the above problems, a modular design of undulating fin based on a cam mechanism is proposed. As shown in Figure 3a, the undulating fin is composed of a tilting socket, a common shaft, cam mechanisms, fin ray units, and a flexible fin.

The undulating fin is clamped on the corresponding cam by eight fin ray units. As shown in Figure 3b, the phase interval between each cam is $\pi/2$ and is installed on the common shaft at an equal distance. As shown in Figure 3c, the fin ray units swing back and forth according to the motion characteristics of the cam and drive the undulating fin to produce sinusoidal motion. If the flexible undulating fin is directly and rigidly connected with the cam, the water-facing angle of the undulating fin will be fixed and result in additional resistance. In order to solve this problem, we put a sliding bearing in the fin ray unit. As shown in Figure 3d, the fin clamp has rotational freedom. When the long fin undulates, the fin surface can adaptively adjust the attack angle related to the local fluid velocity to reduce the additional resistance along the free stream velocity.

The initial geometry of the undulating fin is fan-shaped as shown in Figure 3e. Due to the geometrical constraints of boundary conditions, the sector can form a sinusoidal-shaped conical-banded fin after straightening as shown in Figure 3f. The geometric relationship of the design parameters of the undulating fin can be expressed by the following equations:

$$\begin{cases} R = (d \cdot L_1)/(L_2 - L_1) \\ \alpha = (L_2 - L_1)/(R_2 - R_1) \\ d = R_2 - R_1 \end{cases} \tag{1}$$

where $L_1$ and $L_2$ are the inner arc length and outer arc length of the undulating fin, respectively, which can be approximately calculated by the following formula:

$$
\begin{cases}
L_i = \int_0^{2\lambda} \sqrt{\left(\frac{dx}{dt}\right)^2 + \left(\frac{dy}{dt}\right)^2 + \left(\frac{dz}{dt}\right)^2}\, dt \\
x = t \\
y = R_i \sin(\theta_m \sin(\frac{2\pi t}{\lambda})) \\
z = R_i \cos(\theta_m \sin(\frac{2\pi t}{\lambda}))
\end{cases}
\qquad
\begin{array}{l}
t \in [0, 2\lambda] \\
i = 1, 2
\end{array}
\tag{2}
$$

where $\theta_m$ is the undulating amplitude.

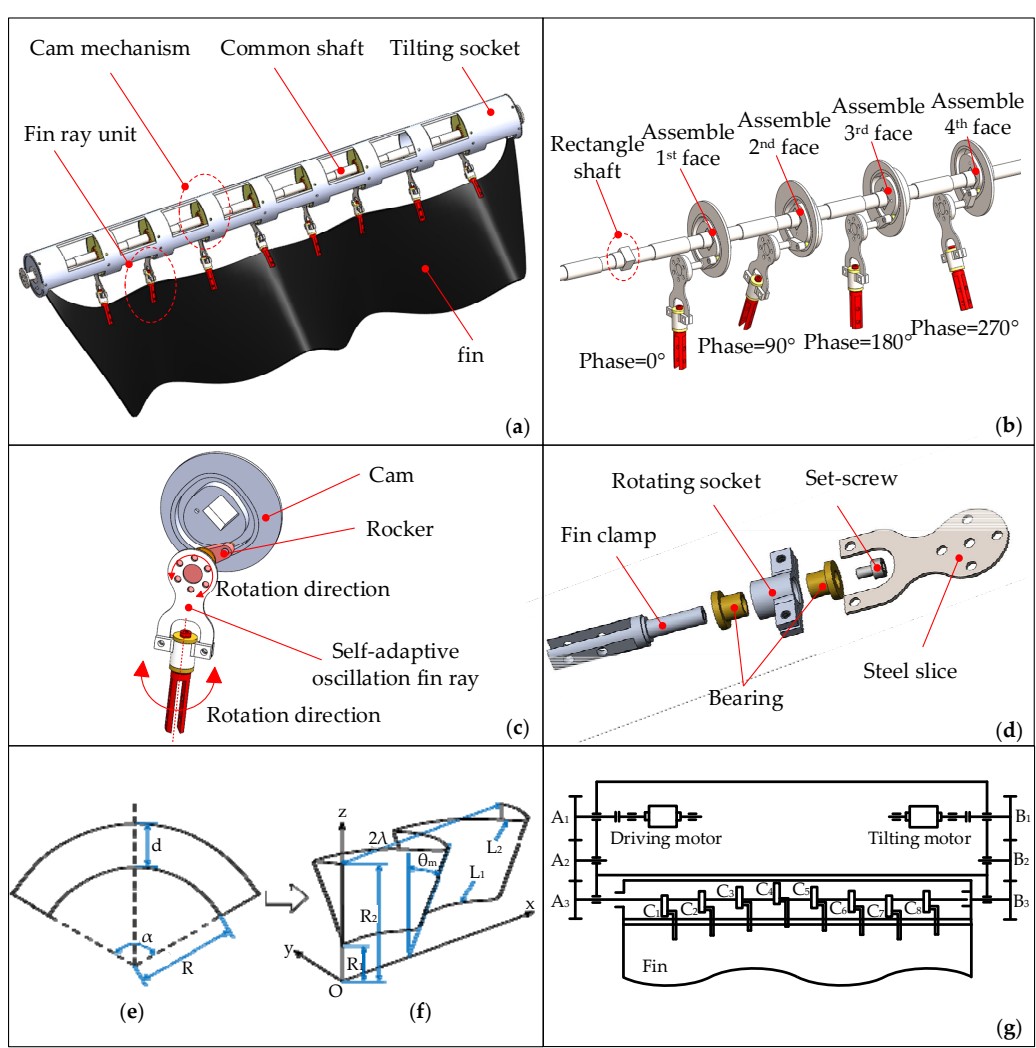

**Figure 3.** Design of the mechanical fin. (**a**) Modular mechanical fin design; (**b**) Phase adjustment mechanism; Four sides of the rectangular shaft correspond to four phases, that is, 0, $\pi/2$, $\pi$, and $3\pi/2$; (**c**) Cam mechanism; (**d**) Self-adaptive oscillation fin ray; (**e**) Initial geometry of the undulating fin; (**f**) Straightened undulating fin; (**g**) Driving and tilting transmission principle of the undulating fin.

The driving and tilting transmission principle of the undulating fin is shown in Figure 3g. The driving motor transmits the rotation to the common shaft through the gear train A1–A2–A3. By changing the motor rotation direction and speed, we can adjust the undulating direction and frequency. Likewise, the tilting motor transmits the rotation motion to the tilting socket through the gear train B1–B2–B3. By controlling the tilting motor, we can change the tilting angle of the undulating fin around the central axis to generate the yaw moment.

## 3. Control Model

### 3.1. Kinematic Model

To describe the motion of the robot, the inertial coordinate system E-XYZ and the robot coordinate system G-XYZ as shown in Figure 4a are established. E is any point, the EZ axis points to the center of the earth, and the EX and EY axes are arbitrarily selected according to the right-hand rule. G is the center of gravity of the robot, the GX axis points to the forward direction of the body, the GZ axis is vertical to the robot, and the GY axis is determined by the right-hand rule.

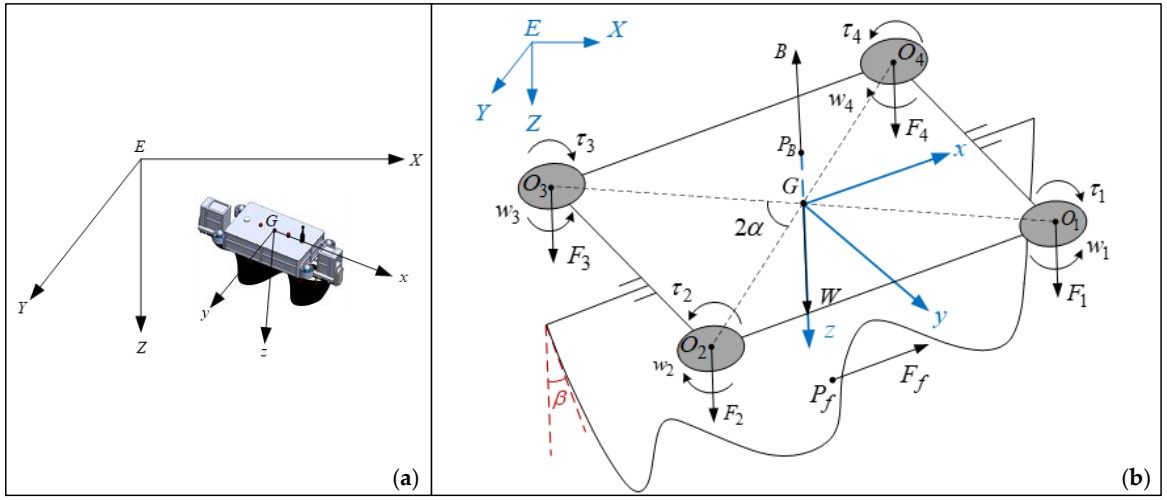

(a)        (b)

**Figure 4.** Kinematics and mechanical model of the robot. (**a**) Inertial coordinate system and robot coordinate system; (**b**) Dynamic model of the robot.

In the inertial coordinate system E-XYZ, the attitude of the robot is $\eta = (\eta_1^T, \eta_2^T)^T$, where $\eta_1 = (X, Y, Z)^T$ is the position of the robot, $\eta_2 = (\phi, \theta, \psi)^T$ is the attitude angle of the robot. The velocity of the robot in the body coordinate system G-XYZ is $v = (v_b^T, w_b^T)^T$, where $v_b = (u_b, v_b, w_b)^T$ is the linear velocity, $\omega_b = (p, q, r)^T$ is the angular velocity.

The kinematic model of the robot can be given by:

$$\dot{\eta} = J(\eta) \cdot v \tag{3}$$

where $J(\eta)$ is the velocity rotation matrix from robot coordinate system to inertial coordinate system, which is defined as:

$$J(\eta) = \begin{pmatrix} J_1(\eta) & 0 \\ 0 & J_2(\eta) \end{pmatrix} \tag{4}$$

$$J_1(\eta) = \begin{pmatrix} c_\psi c_\theta & c_\psi s_\theta s_\phi - s_\psi c_\phi & c_\psi s_\theta c_\phi + s_\psi s_\phi \\ s_\psi c_\theta & s_\psi s_\theta s_\phi + c_\psi c_\phi & s_\psi s_\theta c_\phi - c_\psi s_\phi \\ -s_\theta & c_\theta s_\phi & c_\theta c_\phi \end{pmatrix} \tag{5}$$

$$J_2(\eta) = \begin{pmatrix} 1 & s_\phi t_\theta & c_\phi t_\theta \\ 0 & c_\phi & -s_\phi \\ 0 & s_\phi/c_\theta & c_\phi/c_\theta \end{pmatrix} \tag{6}$$

where $c_{(\cdot)}, s_{(\cdot)}, t_{(\cdot)}$ are the abbreviations of $\cos(\cdot), \sin(\cdot), \tan(\cdot)$, respectively.

### 3.2. Dynamic Model

The robot is subjected to the combined action of gravity, buoyancy, propulsion of propellers, and undulating fin and resistance. As shown in Figure 4b, the weight force of the robot is $F_w$ which acts on point $G$, the buoyancy is $F_B$, and the center of buoyancy is $P_B$. The thrust of propellers and counter torque are $F_i$, $\tau_i$ ($i = 1, 2, 3, 4$), respectively, and

act on $O_i$ ($i$ = 1, 2, 3, 4) which is the centroid of each propeller. The distance between the propellers and the center of the robot is $GO_i = L$ ($i$ = 1, 2, 3, 4) and the angle between the diagonal is $2\alpha$. The equivalent force of the undulating fin is $F_f$ and acts on $P_F(0, 0, z_F)$.

### 3.2.1. Dynamic Model of Propellers

The velocity of the propeller is represented by $w_i$, the thrust and reverse torque are, respectively, represented by $F_i$, $\tau_i$ ($i$ = 1, 2, 3, 4), and the distances between the thrusters are 2a, and 2b, respectively. The thrust and reverse torque of the propeller can be given by:

$$F_i = \text{sign}(w_i)c_T w_i^2, i = 1, 2, 3, 4$$
$$\tau_i = \lambda_i \text{sign}(w_i)c_M w_i^2, i = 1, 2, 3, 4 \tag{7}$$

where $c_T$, $c_M$ are thrust coefficient and reverse torque coefficient, respectively, which can be obtained through the regression polynomials of M's 3 blades propeller open water thrust coefficient diagram [31]; $\text{sign}(\cdot)$ is a symbolic function; $\lambda_i$ is the directional coefficient, determined by the propeller installation direction. In order to counteract the reverse torque in hover, the rotation directions of adjacent propellers are opposite, so it leads to $\lambda_1 = \lambda_3 = 1$, $\lambda_2 = \lambda_4 = -1$.

### 3.2.2. Dynamic Model of Undulating Fin

The undulating fin can generate forward or backward thrust through sinusoidal motion, denoted as $F_f$ and the action point $P_f$ is taken at the center of the fin surface. The center of gravity of the robot is $G$, $GP_f = l$, the tilting angle of the undulating fin is $\beta$, so the coordinate of the action point of equivalent force is $P_f(0, -lc_\beta, ls_\beta)$.

Previous studies have shown that the relationship between the thrust and frequency of undulating fin is a second-order equation:

$$F_f = \text{sign}(w_f)c_f w_f^2 \tag{8}$$

where $w_f$ is the frequency of the undulating fin and $c_f$ is the thrust coefficient of the undulating fin, which can be obtained through simulation and experiment.

### 3.2.3. Model of Resilience

The buoyancy center of the robot is designed to be slightly higher than the center of gravity, so as to provide a certain restoring force and torque to assist the robot to maintain the balance of the body in an emergency. If the position of the center of buoyancy is marked as $P_B(0, 0, z_B)$, and the buoyancy and gravity are, respectively, represented by $B$ and $W$, the resilience matrix acting on the center of gravity can be given by:

$$g(\eta) = \begin{pmatrix} (F_w - F_b)s_\theta \\ (F_b - F_w)c_\theta s_\phi \\ (F_b - F_w)c_\theta s_\phi \\ -z_B F_b c_\theta s_\phi \\ -z_B F_b s_\theta \\ 0 \end{pmatrix} \tag{9}$$

where $\theta$ is the pitch angle of the robot and $\phi$ is the roll angle of the robot.

### 3.2.4. Model of Resistance

The resistance of underwater robots in the process of motion is very complex. At present, the commonly used method is to express the damping as the uncoupled super-

position of primary damping and secondary damping. The damping force matrix can be given by:

$$\boldsymbol{D}(\boldsymbol{v}) = \mathrm{diag}\Big(X_u + X_{u|u|}\big|u, +Y_{v|v|}\big|v\big|, Z_w + Z_{w|w|}\big|w\big|,$$
$$K_p + K_{p|p|}\big|p\big|, M_q + M_{q|q|}\big|q\big|, N_r + N_{r|r|}\big|r\big|\Big) \qquad (10)$$

where $X_u$, $Y_v$, $Z_w$, $K_p$, $M_q$, $N_r$ are the primary damping coefficients in six directions of the body coordinate system, respectively, $X_{u|u|}$, $Y_{v|v|}$, $Z_{w|w|}$, $K_{p|p|}$, $M_{q|q|}$, $N_{r|r|}$ are the secondary damping coefficients in six directions of the body coordinate system, respectively. The damping coefficient can be obtained by CFD.

*3.3. Equation of Motion Control*

We define the control force matrix and 6-DOF control force of the robot as:

$$\boldsymbol{u} = \big(F_1, F_2, F_3, F_4, F_f\big)^T \qquad (11)$$

$$\boldsymbol{\tau} = \big(\tau_X, \tau_Y, \tau_N, \tau_K, \tau_M, \tau_N\big)^T \qquad (12)$$

We define $k = c_M/c_T$ as the proportional coefficient of propellers thrust and reverse torque, the 6-DOF control force can be solved as given by:

$$\boldsymbol{\tau} = \begin{pmatrix} \tau_X \\ \tau_Y \\ \tau_Z \\ \tau_K \\ \tau_M \\ \tau_N \end{pmatrix} = \begin{pmatrix} 0 & 0 & 0 & 0 & 1 \\ 0 & 0 & 0 & 0 & 0 \\ 1 & 1 & 1 & 1 & 0 \\ Ls_\alpha & Ls_\alpha & -Ls_\alpha & -Ls_\alpha & 0 \\ -Lc_\alpha & Lc_\alpha & Lc_\alpha & -Lc_\alpha & lc_\beta \\ k & -k & k & -k & ls_\beta \end{pmatrix} \cdot \begin{pmatrix} F_1 \\ F_2 \\ F_3 \\ F_4 \\ F_f \end{pmatrix} \qquad (13)$$

Under the combined action of robot control force, resistance, and resilience, the dynamic mathematical model of the robot can be given by:

$$\begin{cases} \boldsymbol{\tau} = \boldsymbol{M}(\boldsymbol{v})\dot{\boldsymbol{v}} + \boldsymbol{C}(\boldsymbol{v})\boldsymbol{v} + \boldsymbol{D}(\boldsymbol{v})\boldsymbol{v} + \boldsymbol{g}(\boldsymbol{\eta}) \\ \boldsymbol{\tau} = \boldsymbol{B}\boldsymbol{u} \end{cases} \qquad (14)$$

The meanings of physical quantities in the Equation (14) are as follows:

$\boldsymbol{\tau}$-6 × 1 control matrix represents the 6-DOF force and moment and the component expression is $\boldsymbol{\tau} = \big(\tau_X, \tau_Y, \tau_N, \tau_K, \tau_M, \tau_N\big)^T$;

$\boldsymbol{M}(\boldsymbol{v})$-6 × 6 inertia matrix;

$\boldsymbol{D}(\boldsymbol{v})$-6 × 6 resistance matrix;

$\boldsymbol{C}(\boldsymbol{v})$-6 × 6 centrifugal force and Coriolis force matrix;

$\boldsymbol{g}(\boldsymbol{\eta})$-6 × 6 resilience matrix;

$\boldsymbol{u}$-5 × 1 input force matrix represents the thrust of the propeller and undulating fin;

$\boldsymbol{B}$-6 × 5 control matrix represents the conversion relationship between the input force and the 6-DOF force of the robot.

As the robot navigates at low speed, the assumptions can be made as follows:

1.  Ignore the additional mass force [32], i.e., $\boldsymbol{M}(\boldsymbol{v}) = \mathrm{diag}\big(m, m, m, J_{xx}, J_{yy}, J_{zz}\big)$;
2.  Assume that Coriolis force and centrifugal force have negligible influence on the motion [32], i.e., $\boldsymbol{C}(\boldsymbol{v}) = 0$;
3.  According to the dynamic analysis, the control component of the robot along the *Y*-axis direction is 0, so the sway motion along *Y*-axis is ignored.

Through the above analysis, the control force matrix, mass matrix, resistance matrix, and resilience matrix are substituted into Equation (12) and expanded. The motion is decomposed into independent channels, so the 6-DOF motion dynamics equation is:

$$
\begin{cases}
m\dot{u} = \tau_X - (X_u + X_{u|u|}|u|)u - (F_w - F_b)s_\theta \\
m\dot{v} = 0 \\
m\dot{w} = \tau_Z - (Z_w + Z_{w|w|}|w|)w - (F_b - F_w)c_\theta c_\phi \\
J_{xx}\dot{p} = \tau_K - (K_p + K_{p|p|}|p|)p + z_B F_b c_\theta s_\phi \\
J_{yy}\dot{q} = \tau_M - (M_q + M_{q|q|}|q|)q + z_B F_b s_\theta \\
J_{zz}\dot{r} = \tau_N - (N_r + N_{r|r|}|r|)r
\end{cases}
\tag{15}
$$

Through the kinematic matrix conversion, the linear and angular acceleration of the robot in the geographical coordinate system can be given by:

$$
\begin{pmatrix} \ddot{X} \\ \ddot{Y} \\ \ddot{Z} \end{pmatrix} =
\begin{pmatrix}
c_\theta c_\psi & s_\theta c_\psi s_\phi - s_\psi c_\phi & s_\theta c_\psi c_\phi + s_\psi s_\phi \\
c_\theta s_\psi & s_\theta s_\psi s_\phi + c_\psi c_\phi & s_\theta s_\psi c_\phi - c_\psi c_\phi \\
-s_\theta & c_\theta s_\phi & c_\theta c_\phi
\end{pmatrix}
\cdot
\begin{pmatrix} \dot{u} \\ \dot{v} \\ \dot{w} \end{pmatrix}
\tag{16}
$$

$$
\begin{pmatrix} \ddot{\phi} \\ \ddot{\theta} \\ \ddot{\psi} \end{pmatrix} =
\begin{pmatrix}
1 & s_\phi t_\theta & c_\phi t_\theta \\
0 & c_\phi & -s_\phi \\
0 & s_\phi/c_\theta & c_\phi/c_\theta
\end{pmatrix}
\cdot
\begin{pmatrix} \dot{p} \\ \dot{q} \\ \dot{r} \end{pmatrix}
\tag{17}
$$

*3.4. Control Model and Strategy*

During the movement of the underwater robot, due to the nonlinear and time-varying characteristics of the external water flow disturbance, the system is prone to oscillation and even instability. So, it is difficult to establish an accurate hydrodynamic model of the robot. To solve the case, we propose a 4-DOF cascade PID controller. The control schematic diagram of the system is shown in Figure 5. For the three-axis angle and Z-axis position of the robot, an independent cascade PID closed-loop controller was designed, and speed feedback introduced to improve the response speed of the system. The speed open-loop control is adopted for the surge motion of the robot because we will conduct experimental research on the velocity of the robot with high-frequency undulation. The reference input of the control system is $R = (Z_d, \phi_d, \theta_d, \psi_d, w_d)^T$, the components represent the expected value of depth and triaxial angle and the undulating frequency, respectively. The output of the controller is $U_d = (\tau_{Zd}, \tau_{Kd}, \tau_{Md}, \tau_{Nd})^T$, the components represent the expected vertical force, rolling torque, pitching torque, and yaw torque calculated by the four-channel controllers. To Define a control parameter matrix $U_o = (w_1, w_2, w_3, w_4, w_f, \beta)^T$, the components are the rotational speed of the four propellers, the undulating frequency, and the tilting angle, which are the final parameters directly controlled and the output through the motor.

In order to convert the expected force and torque calculated by the controller into direct control parameters, a control distributor is introduced. We quote a function to describe it, i.e., $U_o = f(U_d)$. It is too complicated to obtain the specific expression of the mapping relationship directly from the control force expression, so we use a decomposition strategy to decompose the system into the quadrotor subsystem and the undulating fin subsystem.

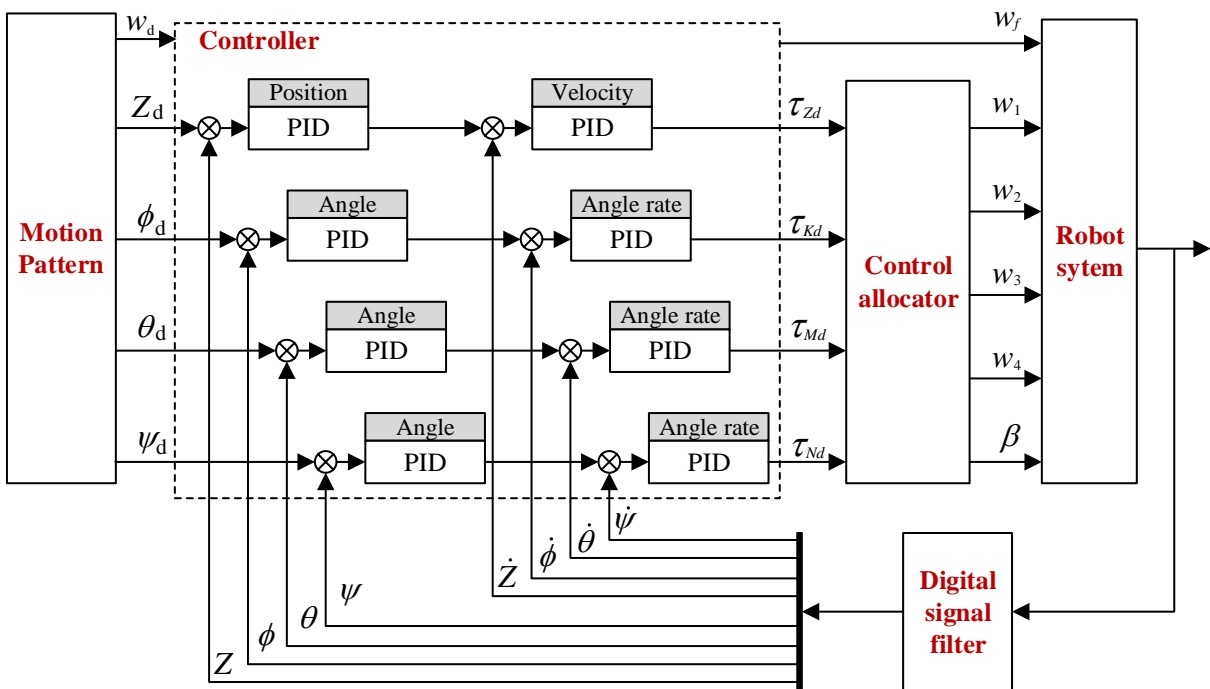

**Figure 5.** Cascade PID controller of depth and attitude.

The quadrotor subsystem can be given by:

$$
\begin{pmatrix} \tau_{Zd} \\ \tau_{Kd} \\ \tau_{Md} \\ \tau_{Nd} \end{pmatrix} = \begin{pmatrix} 1 & 1 & 1 & 1 \\ a & a & -a & -a \\ -b & b & b & -b \\ k & -k & k & -k \end{pmatrix} \cdot \begin{pmatrix} F_1 \\ F_2 \\ F_3 \\ F_4 \end{pmatrix}
\tag{18}
$$

The inverse of the matrix can be obtained:

$$
\begin{pmatrix} F_1 \\ F_2 \\ F_3 \\ F_4 \end{pmatrix} = \frac{1}{4} \begin{pmatrix} 1 & 1 & -1 & 1 \\ 1 & 1 & 1 & -1 \\ 1 & -1 & 1 & 1 \\ 1 & -1 & -1 & -1 \end{pmatrix} \cdot \begin{pmatrix} \tau_{Zd} \\ \tau_{Kd}/a \\ \tau_{Md}/b \\ \tau_{Nd}/k \end{pmatrix}
\tag{19}
$$

We know that the relationship between the thrust of the propeller and the rotation frequency is quadratic, and the thrust coefficient and the reverse torque coefficient can be obtained through experiments. Then, the specific expression of the propeller speed can be obtained, but this is not necessary because the PID controller can compensate for these unknown parameters. Using pseudo control quantity, the propeller speed can be obtained as:

$$
\begin{pmatrix} w_1 \\ w_2 \\ w_3 \\ w_4 \end{pmatrix} = \frac{1}{4} \begin{pmatrix} 1 & 1 & -1 & 1 \\ 1 & 1 & 1 & -1 \\ 1 & -1 & 1 & 1 \\ 1 & -1 & -1 & -1 \end{pmatrix} \cdot \begin{pmatrix} \tau_{Zd} \\ \tau_{Kd} \\ \tau_{Md} \\ \tau_{Nd} \end{pmatrix}
\tag{20}
$$

The undulating fin subsystem can be given by:

$$
\begin{pmatrix} \tau_Z \\ \tau_K \\ \tau_M \\ \tau_N \end{pmatrix} = \begin{pmatrix} 0 & \cos\beta \\ 0 & 0 \\ l\cos\beta & 0 \\ l\sin\beta & 0 \end{pmatrix} \cdot \begin{pmatrix} F_u \\ F_w \end{pmatrix}
\tag{21}
$$

Since the undulating frequency is directly given, $F_u$, $F_w$ can be obtained. The problem is how to get the tilting angle of the undulating fin. In this case, we adopt a proportional control distribution strategy as shown in Equation (22):

$$\begin{cases} w_f = w_d \\ \beta = \lambda \tau_{Nd}, \beta \in [-60°, 60°] \end{cases} \tag{22}$$

where $\lambda$ can modify and determine the speed of steering.

## 4. Simulation and Experiment

### 4.1. Prototype and Experimental Environment

The prototype, ground control station, and test water tank in this paper are shown in Figure 6. The detailed parameters of the robot are shown in Table A1, the pool is 4 m long, 2 m wide, and 2 m high. As in most cases, the wave and wind are very weak, so the underwater environment can be treated as calm water.

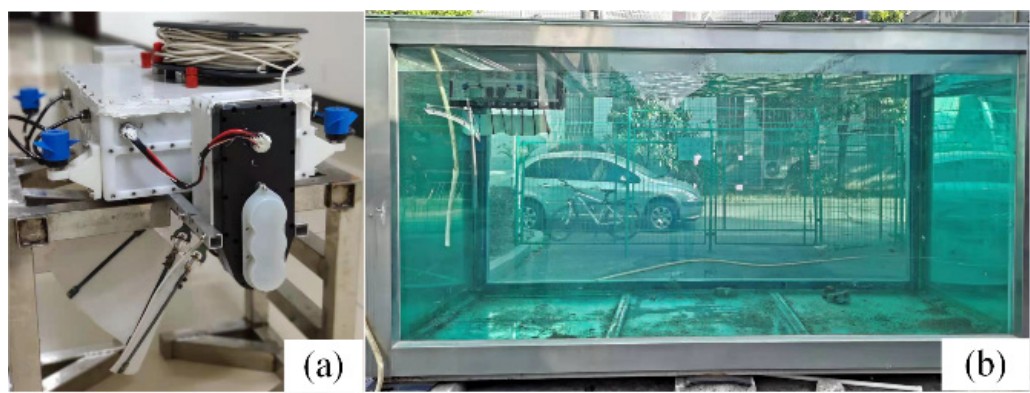

**Figure 6.** Prototype of the robot and experimental tank. (**a**) Prototype; (**b**) Experimental tank.

The control mode of the robot is that the remote controller control and the upper computer control are carried out at the same time, and the upper computer has a higher priority. Using the upper computer software independently developed based on Qt, the process is as follows: first, send the desired command to the robot through the upper computer; then, the airborne control system controls the robot to move after receiving the command; and, finally, the airborne control system uploads the real-time status of the robot to the upper computer for data storage.

### 4.2. Simulation Parameters and Models

According to the model of resistance, control force, and the resilience obtained from the numerical simulation of fluid dynamics, the 6-DOF dynamic equation of the robot can be obtained, which provides a basis for the design of an independent controller.

As shown in Figure 7a,b, the numerical simulation model of the underwater robot is established in Fluent. The grid of the inner basin is encrypted and the total length and height of the outer basin are set to 10 times the length of the inner basin to simulate the motion of the robot in the forward and vertical directions. Considering the calculation time and accuracy, the number of grids selected is 983,910. Define_Grid_Motion is used to define and control the motion of the flexible undulating fin's underwater motion simulation. For the generation of dynamic meshes, the diffusion method and local cell remeshing are used to continuously re-divide the mesh.

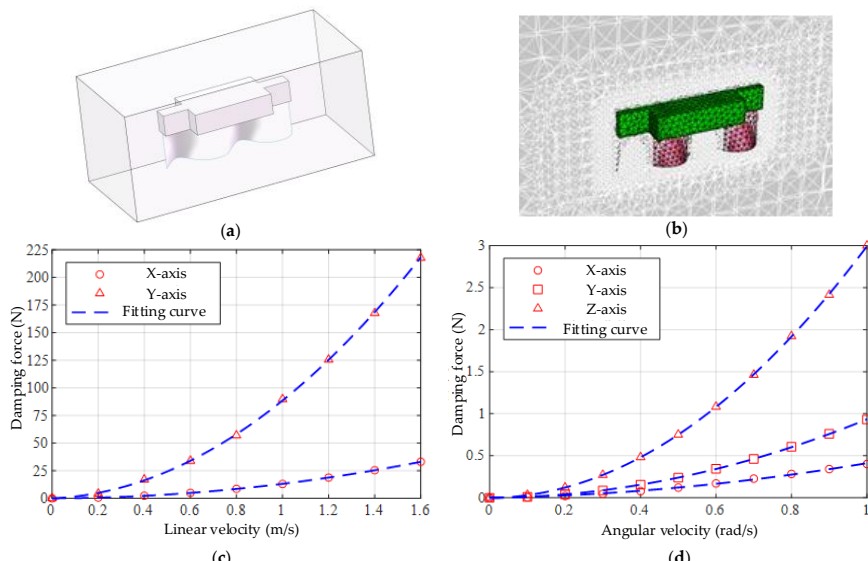

**Figure 7.** Numerical simulation analysis of damping coefficient. (**a**,**b**) FEM of the robot; (**c**) Resistance–velocity curve of uniaxial linear motion by CFD simulation; (**d**) Moment of the resistance–velocity curve of uniaxial rotational motion by CFD simulation.

To begin, we simulate the damping force/torque of the robot at different velocities when moving in a single degree of freedom and then identify the model parameters by the least square method. After that, we can obtain the damping coefficient of heave, surge, roll, pitch, and yaw motion. Figure 7c shows the numerical simulation results and fitting curve of damping force velocity change under linear motion. Figure 7d shows the numerical simulation results and fitting curve of damping force angular velocity change under rotating motion. The identification results are shown in Table 1:

**Table 1.** Result of damping coefficient by CFD simulation.

| Primary Damping Coefficient | | Secondary Damping Coefficient | |
|---|---|---|---|
| $X_u$ | 0.7164 | $X_{u\|u\|}$ | 12.4744 |
| $Y_v$ | - | $Y_{v\|v\|}$ | - |
| $Z_w$ | 10.9893 | $Z_{w\|w\|}$ | 78.8397 |
| $K_p$ | 0.0760 | $K_{p\|p\|}$ | 0.3359 |
| $M_q$ | 0.0264 | $M_{q\|q\|}$ | 0.9075 |
| $N_r$ | 0.0134 | $N_{r\|r\|}$ | 2.9787 |

In order to verify the effect of the controller and simulate the influence of PID parameters on stability control, Matlab/Simulink is used to design the simulation model of the control system. The schematic diagram of the motion control simulation is shown in Figure 8. The reference input of the control system is $R = (Z_d, \phi_d, \theta_d, \psi_d, w_d)^T$, the components represent the expected value of depth and triaxial angle and the undulating frequency, respectively. The initial state is $s = (Z_0, \phi_0, \theta_0, \psi_0, w_0)^T$. By setting different reference inputs and initial conditions, the maneuverability and attitude stability of the robot in typical motion modes such as heave motion, surge motion, and in-situ steering motion are discussed.

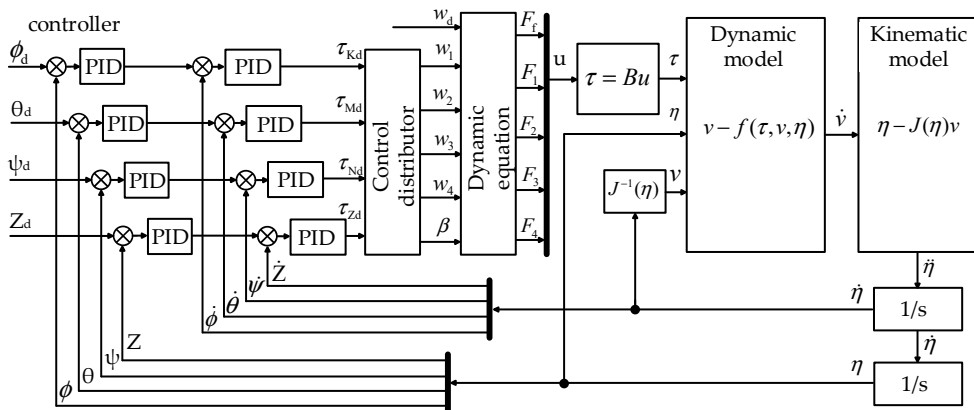

**Figure 8.** Schematic diagram of motion control simulation.

### 4.3. Simulation and Experiment Results

In order to verify the design and control model of the robot, we adopt the method of combining simulation with prototype experiment to verify the motion of surge, heave, and yaw. In particular, we tested the propulsion of the undulating fin under different frequencies.

The experimental results are saved by image and data. On the one hand, we set a high-speed camera (GoPro) in front of the pool to capture the experimental images of the robot in each group of experiments. On the other hand, the robot uploads real-time data at the frequency of 50 Hz, including attitude angle, depth, battery capacity, and other data. The laptop displays the data through the self-developed upper computer and saves the data in the file (Supplementary Materials).

#### 4.3.1. Heave Motion

In the heave motion simulation, the initial state of the robot is set to $s = (0,0,0,0,0)^T$, the reference input is set to $R = (1,0,0,0,0)^T$ ($t < 20$ s) and $R = (0,0,0,0,0)^T$ ($t > 20$ s). The simulation results are shown in Figure 9a, the robot can reach the desired depth quickly, and the error of overshoot and steady state is small. At the same time, the robot can adjust the attitude angle to stabilize at 0° in the case of noise interference.

In the heave motion experiment, we first place the robot on the water surface and then issue a desired depth value to let the robot track to this position. After the robot hovers stably, we reset the desired depth to 0 m to make it move back to the surface. Meanwhile, the depth and attitude angle of the robot are recorded by the upper computer. Due to the error of the analog sensor in the simulation, random noise interference with the frequency of 10 Hz and amplitude of 0.02 m and 2° is applied to the depth and triaxial angle, respectively.

The time sequence of the heave motion experiment is shown in Figure 9c and the curve of robot depth and three-axis attitude angle with time is shown in Figure 9b. It can be seen that the robot quickly responds to the set desired depth and, finally, stably hovers with a maximum tracking velocity of about 0.23 m/s, with an overshoot of about 12%, and a steady-state error of ±0.02 m. It can be seen from the attitude angle curve in the lower figure of Figure 9b that the angle fluctuation is accompanied by the descending process of the robot, but it can be adjusted rapidly. The angle fluctuation range is less than ±2° in the steady state.

The experimental results of heave movement show:

- The robot can effectively realize the closed-loop control of depth, and the sinking and floating speed can reach 0.27 m/s;
- The heave motion of the robot is independent, the robot has no surge motion, sway motion, and tilt of attitude angle, which indicates that the robot has achieved independent motion along the Z-axis;

- The robot has good motion stability, the roll, pitch, and yaw angle of the robot remain stable during the process of descent, hovering, and floating.

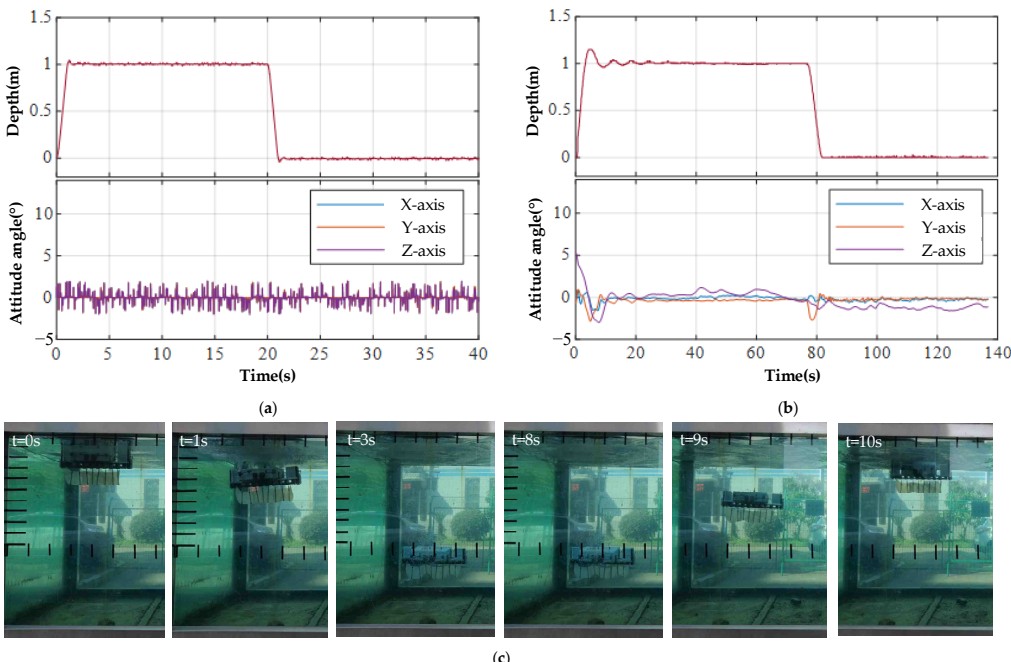

**Figure 9.** Results of heaving motion. (**a**) Simulation result of heaving motion; (**b**) Experiment result of heaving motion; (**c**) Time sequence of heave motion.

### 4.3.2. Yaw Motion

In the yaw motion simulation, the initial state of the robot is set to $s = (0, 0, 0, 0, 0)^T$, and the reference input is set to $R = (0, 0, 0, 90°, 0)^T$. The simulation results are shown in Figure 10a, the robot quickly tracks to the given heading angle. The roll angle and pitch angle are stable in the horizontal position while the depth is maintained at 0 m.

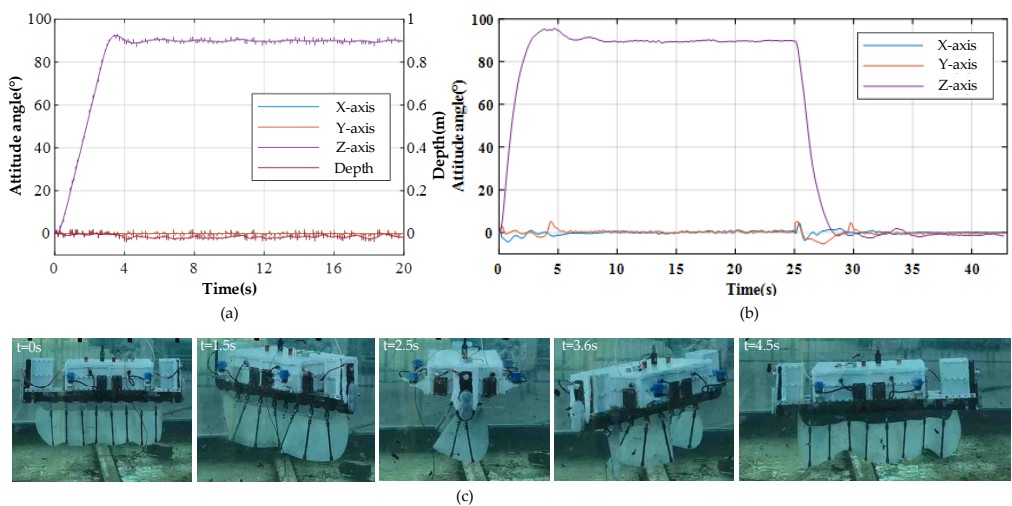

**Figure 10.** Results of yaw motion. (**a**) Simulation result of yaw motion; (**b**) Experiment result of yaw motion; (**c**) Time sequence of yaw motion.

In the yaw motion experiment, the undulating fin does not work and only uses the propellers to adjust the yaw angle. First, the robot is suspended in the center of the tank (L = 1 m) and the yaw angle is initialized to 0°. Then, we issue the command of yaw angle

pointing at 90° to observe the response of the robot under the step signal. After the robot is stable, we issue the command of yaw angle pointing at 0° in order to return it to its initial position.

Figure 10b shows the time-varying curve of the attitude angle of the robot in the yaw motion. It can be seen that the heading angle has been tracked to the specified angle in the front and back stages.

Figure 10c shows the image sequence in the yaw motion. It can be seen that the robot rotates 90° around the Z-axis in 2.5 s and 180° around the Z-axis in 4.5 s. By comparing the black marks on the tank, it can be seen that the depth of the robot has hardly changed, and the robot remains balanced. It shows that the yaw motion is also an independent motion and the robot only rotates around the Z-axis.

### 4.3.3. Steering Motion

In the Steering motion experiment, we place the robot in the corner of the tank. Then we tilt the undulating fin 60° to one side and issue a forward command to make the robot fin undulate at the frequency of 1 Hz. Figure 11a shows the image sequence of the steering-left motion. It can be seen that the undulating fin is no longer centered, which results in a yaw moment. The robot steers left under the action of the yaw moment and the maximum steering angular velocity of the robot is about 30°/s.

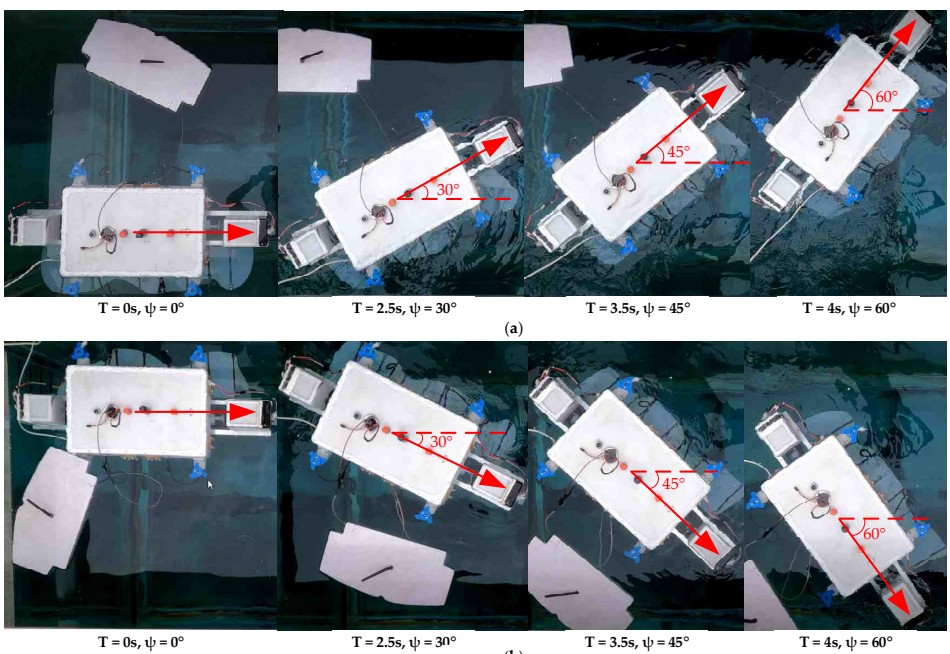

**Figure 11.** Results of steering motion. (**a**) Time sequence in steering-left motion experiment, in which case the undulating fin was tilted to the right side; (**b**) Time sequence in steering-right motion experiment, in which case the undulating fin was tilted to the left side.

Similar to the steering-left experiment, we tilt the undulating fin 60° to the right side in the steering-right experiment. Then, we issue a forward command to make the robot fin undulate forward at the frequency of 1 Hz. Figure 11b shows that the robot steers right successfully. The results of the traveling steering experiment show that the surge angle of the robot can be effectively adjusted only by the tilting angle of the undulating fin.

### 4.3.4. Surge Motion

The most important motion of an underwater robot is surge motion. Due to the complex resistance in surge motion, the robot is easy to yaw or overturn under the force

of the water. To solve this problem, we control the robot to maintain a fixed depth and heading angle through real-time adjustment by four propellers.

In the surge motion simulation, the initial state of the robot is set to $s = (0, 0, 0, 0, 0)^T$, and the reference input is set to $R = (1, 0, 0, 0, 100)^T$. We make the robot move from the surface to a 1 m depth, then make the fin undulate at the frequency of 2 Hz (100 rpm of the motor) at the same time. The simulation result is shown in Figure 12a. While tracking to the desired depth, the robot accelerates to the maximum velocity. There is no sway motion in the experiment. The attitude angle and depth are stable as reference input values, which is consistent with the theoretical analysis result.

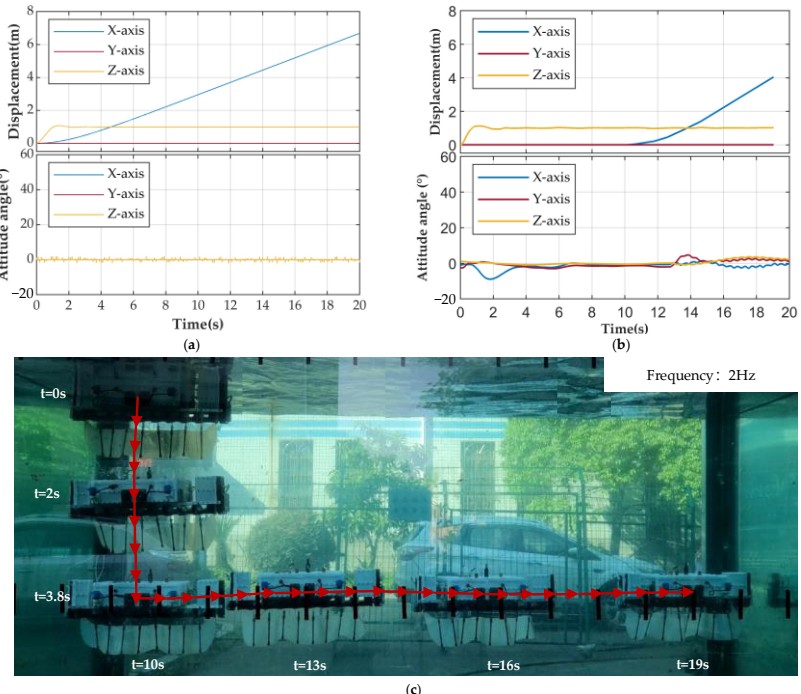

**Figure 12.** Results of surge motion. (**a**) Simulation result of surge motion; (**b**) Experiment result of surge motion; (**c**) Time sequence of surge motion.

In the surge motion experiment, we made the robot move forward and backward on the surface and underwater at a depth of 1 m, respectively. First, we placed the robot at the left end of the tank and suspended it on the surface of the water, then issued a diving command to move it to a depth of 1 m. After that, we controlled the robot to surge forward at a 2 Hz undulating frequency, braked when moving to the right end of the tank, and ended by surging backward to the left end of the tank. Figure 12b shows the depth and attitude angle versus time in the surge motion experiment, and Figure 12c shows the time sequence in the surge motion of the robot. From the experimental image and diagram, it can be seen that the robot first descends to a specified depth and then surges forward. The angle fluctuation range is less than $\pm 5°$ in the lower figure of Figure 12. So, the robot keeps its balance and surges forward at the fixed depth during the motion.

The surge motion experiment shows:

- The robot can achieve attitude stability in surge movement, and the angle fluctuation range is less than $\pm 5°$;
- The depth closed-loop controller and attitude angle closed-loop controller play an effective role in regulation;
- The surge motion of the robot is independent and does not accompany the motion of the other five degrees of freedom.

### 4.3.5. Velocity Experiment

In the last group of experiments, we measured the propulsive velocity of the robot. First, the robot is placed at the left end of the tank and floats on the surface. We took the frequencies of the undulating fin $f = 1.0 - 6.0$ Hz every 0.5 Hz so that we could conduct 11 sets of experiments. First, the laptop controls the robot to surge forward at the specified frequency and brake when it reaches the right end of the tank, we then return it to its initial position. We record the time that the robot passes through the middle of the tank (l = 0.1–0.3 m) three times at each frequency, then calculate the average velocity.

The experimental result is shown in Figure 13b, and it can be seen that with the increase of undulating frequency, the propulsive velocity of the robot also increases, and the trend is consistent with the simulation results. When the undulating frequency is 6 Hz, the propulsive velocity of the robot reaches 1.2 m/s (1.5 BL/s). A comparison of the existing work is shown in Table 2, which shows that the performance is good since it is well acknowledged that the long fin is not optimized for high speed. Compared with the CFD simulation results in Figure 13a, the actual velocity–frequency curve trend of the prototype is consistent with the simulation curve. The propulsive velocity of the actual robot is not as fast as the simulation one. Two reasons were speculated. On the one hand, the fluid domain is simplified in the simulation model, and the fluid resistance of the actual robot prototype is greater than that of the simulation model. On the other hand, the waveform generated by the undulating fin formed by eight rigid fin rays is not an ideal sine wave. The waveform is more distorted especially under high-speed undulation, thus, reducing the propulsion efficiency. In addition, due to the limitation of the pool length, when $f > 3$ Hz, the robot may have braked before accelerating to the maximum speed, resulting in the calculated average velocity being low. In the future, the experiments will be carried out in a larger field, the high-precision accelerometers will be equipped for the robot, and the velocity measurement method will be improved.

**Table 2.** Characteristics of existing work.

| Name | Type | Velocity (m/s) | Velocity (BL/s) | Frequency (Hz) |
|---|---|---|---|---|
| *Carangiform* swimming robot [9] | BCF | 1.76 | 2.2 | 3.5 |
| *Ostraciiform* underwater robot [12] | BCF | 0.3 | 2 | 14 |
| Single motor actuated robotic fish [33] | BCF | 1.14 | 3.08 | 9 |
| Wire-driven robotic fish [34] | BCF | 0.08 | 0.25 | 1 |
| Four link robotic fish [35] | BCF | 0.32 | 0.8 | 2 |
| EMA robotic fish [36] | BCF | 0.03 | 0.33 | 3.7 |
| Robo-ray II [8] | MPF | 0.16 | 0.5 | 1.2 |
| Robotic stingray [37] | MPF | 0.04 | 0.18 | 0.5 |
| Undulatory Fin Prototype [38] | MPF | 0.67 | 1.45 | 1.5 |
| **Robot in this paper** | **MPF** | **1.2** | **1.5** | **6** |

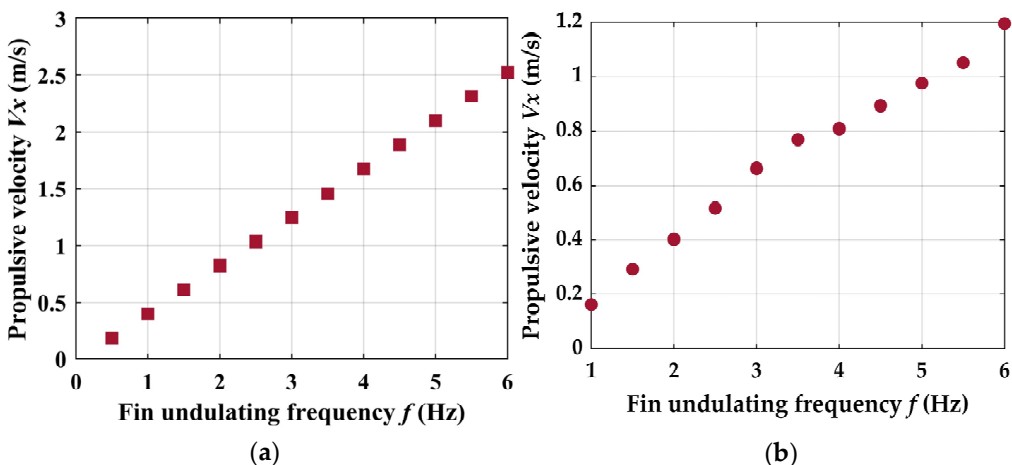

**Figure 13.** The (**a**) Simulation and (**b**) Experiment results of propulsive velocity ($V_x$) versus undulating fre-quency ($f$).

## 5. Discussion

In this paper, we developed an underwater robot based on the hybrid propulsion of a quadrotor and a bionic undulating fin. In this section, the performance of the bionic robot is further discussed in terms of maneuverability, stability, control strategy, and applicability.

### 5.1. Performance of Maneuverability

The undulating fin inspired by *Gymnarchus niloticus* plays an important role in enabling the underwater robot to realize the basic motion of heave, yaw, steering, and surge. As mentioned in Section 2, the bionic fin has two features. First, driven by the cam mechanism, the long fin can undulate at a high frequency (about 6 Hz). The fact that the propulsive velocity of the robot can reach 1.2 m/s (1.5 BL/s) shows that the robot can swim fast (relative to other MPF underwater robots) even though the *Gymnarchus niloticus* in nature cannot achieve this kind of speed. The second feature is that the long fin can tilt to the left/right side. The steering motion experiment shows that the yaw angle of the robot can be effectively adjusted only by tilting the angle of the undulating fin. From a design point of view, because the cam mechanism can theoretically achieve arbitrary regular oscillation, therefore, compared with the crank rocker mechanism [23] and the crank slider mechanism [39], the cam mechanism can provide more design possibilities, such as an undulating fin with a sine wave or cycloid wave.

### 5.2. Performance of Stability

Considering the excellent performance and mature control scheme of the quadrotor in attitude stability, we chose it for supplementary propulsion to achieve robot hover. So, the quadrotor is the other vital component of the underwater robot that hovers the robot and assists in maintaining its balance. The simulation and experiment results show that the attitude angle fluctuation range is less than $\pm5°$ in basic motion and hovering. This means that the robot can achieve attitude stability which fulfills our design purpose. The use of quadrotors sacrifices hydrodynamic performance in the direction of the robot's wave-fin propulsion, but also increases the robot's hovering function and maneuverability in other directions. In particular, when the wave fin fails, the quadrotor can help the robot to complete the task, thereby improving the robot's motion reliability.

### 5.3. Control Strategy

In practical control, we must consider the disturbances within the environment. Even in the limited experimental tank, the robot may be rotated by some uncertain disturbances induced by itself. Taking the surge motion as an example, during the surge, the robot should move straight forward along the *X*-axis and the propellers should work for the

attitude adjustment. If the error between the detected roll angle and the initial one is larger than 5°, which means an unnecessary rotation clockwise, then the propulsive force of the propellers at the right side of the surge orientation will be increased to a larger amplitude to generate an added resistance torque compensating the unnecessary rotation torque. This kind of force compensation and attitude adjustment are also working in the control process of pitch angle, yaw angle, and depth during the multi-mode locomotion. On the other hand, as the six-degree-of-freedom force generated by the four propellers is coupled, it is also a challenge to accurately allocate the thrust to the four propellers based on the calculation results of the controller. In this paper, a 4-DOF cascade PID controller and its control allocation algorithm were designed for this propulsive force adjusting. We conducted simulations and experiments to verify the effect of the controller and test the influence of PID parameters on stability control. The simulation and experimental results clearly show that the robot is able to realize heave motion, surge motion, steering motion, and maintain a stable hover.

*5.4. Applicability*

Owning to the aforementioned advantages, the novel underwater robot is suitable for a variety of tasks such as marine exploration, rescue, and operation. The most common example of such a task in the oil and gas industry is using the robot to make detailed maps of the seafloor before building a subsea infrastructure. In addition, the robot can carry the camera and robotic arm for underwater operations such as pipe maintenance and repair.

The present underwater robot prototype has successfully realized basic motion and high-frequency propulsion. However, compared to our design purpose and the simulation results, its swimming performance still has room for improvement. We summarized future work for the underwater robot as follows: first, the shape of the robot can be designed to be a fish-like shape to improve the hydrodynamic performance of the robot in the directions of moving forward, rising, and sinking; second, the cam mechanism can be optimized by replacing the rigid shaft with a flexible shaft and improving the structure of the fin with soft material to generate an ideal sine wave; and, last but not least, the control strategy can be improved to enhance stability and maneuverability.

**6. Conclusions**

In this paper, we have presented a novel underwater robot based on the hybrid propulsion of a quadrotor and undulating fin. This robot is suitable for replacing or accompanying human beings to execute tasks such as marine exploration, rescue, and operation. We conclude that an underwater robot that is modular designed and based on a hybrid propulsion of quadrotor and undulating fin, and with a cam mechanism used to drive the entire undulating fin through a motor, which also reduces control difficulties while improving the undulating frequency to 6 Hz, is better than other concepts with the same functionality. Next, the 6-DOF dynamic model of the robot was established and the multi-mode motion control of the robot was realized based on a 4-DOF cascade PID controller. The power distribution strategy of the robot in different motion modes was proposed. The simulation and hardware experimental results show that the robot can realize heave motion, surge motion, turn in place motion, and maintain a stable hover in narrow spaces. In hardware experiments, when the undulating frequency is 6 Hz, the maximum propulsive velocity of the robot can reach 1.2 m/s (1.5 BL/s). Therefore, the robot has great potential for underwater operation and has good maneuverability in narrow spaces.

**Supplementary Materials:** The supporting information can be downloaded at https://www.mdpi.com/article/10.3390/jmse10091327/s1, Video S1: prototype test.

**Author Contributions:** Conceptualization, X.Z. and Z.L.; Software, X.Z., Y.X. and M.X.; Data curation, X.Z. and Q.Y.; Writing—original draft preparation, X.Z. and Z.L.; Writing—review and editing, Z.L. and J.S. All authors have read and agreed to the published version of the manuscript.

**Funding:** This research was funded by the National Natural Science Foundation of China, grant number 52,105,289.

**Institutional Review Board Statement:** Not applicable.

**Informed Consent Statement:** Not applicable.

**Data Availability Statement:** The data used to support the findings of this study are available from the corresponding author upon request.

**Conflicts of Interest:** The authors declare no conflict of interest.

## Appendix A

**Table A1.** System parameters of the underwater robot.

| | | |
|---|---|---|
| | **Mass** $m$ | 15.02 kg |
| | **Y-axis inertia** $J_{xx}$ | $0.119 \text{ kg} \cdot \text{m}^2$ |
| **Mass parameter** | **Y-axis inertia** $J_{yy}$ | $0.592 \text{ kg} \cdot \text{m}^2$ |
| | **Z-axis inertia** $J_{zz}$ | $0.563 \text{ kg} \cdot \text{m}^2$ |
| | **No-load buoyancy** $B$ | 160 |
| | **Buoyant center** $z_B$ | $-0.04$ m |
| | **Distance between the propellers** $L$ | 0.281 m |
| | **Angle between the diago-nal** $\alpha$ | $23.6°$ |
| **Propeller parameter** | **Maximum rotating speed** $w_{\max}$ | 10,000 rpm |
| | **Thrust coefficient** $c_T$ | $2.188 \times 10^{-7}$ |
| | **Reverse torque coefficient** $c_M$ | $1.944 \times 10^{-9}$ |
| | **Fin width** $d$ | 0.143 m |
| | **Flexible arc angle** $\alpha$ | $\pi/3$ |
| | **Arc inner diameter** $R$ | 0.8 m |
| | **Wave length** $\lambda$ | 0.419 m |
| | **undulating amplitude** $\lambda$ | 0.06 m |
| | **Fin coefficient** $c_f$ | $6.276 \times 10^{-4}$ |
| **Fin parameter** | **Fin center** $z_F$ | 0.16 m |
| | **Maximum undulating fre-quency** $f_{\max}$ | 6 Hz |

Substituting the system parameters in Tables 1 and A1 into Equation (15), the mathematical model of the robot is as follows:

1. The model of surge motion: $\dot{u} = 0.067\tau_X - (0.048 + 0.832|u|)u + 0.653s_\theta$ ;
2. The model of heave motion: $\dot{w} = 0.067\tau_Z - (0.733 + 5.256|w|)w - 0.653c_\theta c_\phi$ ;
3. The model of roll motion: $\dot{p} = 8.403\tau_K - (0.639 + 2.823|p|)p - 52.7c_\theta s_\phi$ ;
4. The model of pitch motion: $\dot{q} = 1.689\tau_M - (0.045 + 1.533|q|)q - 52.7s_\theta$ ; The model of yaw motion: $\dot{r} = 1.776\tau_N - (0.024 + 5.291|r|)r$ .

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
