# Peer review of "Design and Control of an Underwater Robot Based on Hybrid Propulsion of Quadrotor and Bionic Undulating Fin"

_jmse, doi:10.3390/jmse10091327_

Round 1
Reviewer 1 Report
This work presents an underwater vehicle with a hybrid propulsion system. The overall structure of the paper is fine. The title is appropriate and matches the content of the paper. The abstract is clear, organized and well written. The introduction provides sufficient background and includes relevant references. Even though a few corrections need to be addressed throughout the paper, the English is acceptable, and the ideas are clearly transmitted to the reader. References are recent, relevant and appropriate in number. I have a few suggestions to further improve the overall quality of the paper.
Page 1. The fish of the BCF mode twisted its body for producing the force of the forward propulsion.
I think "bend" or "undulate" is a better way to describe BCF locomotion
Page 2. A few works could be added to the state of the art for the sake of completeness:
About BCF Locomotion
Costa D., Palmieri G., Palpacelli M. C., Scaradozzi D., Callegari M., Design of a Carangiform Swimming Robot through a Multiphysics Simulation Environment. Biomimetics, Vol. 5, No. 4, pp. 46, 2020
About hybrid propulsion systems
Scaradozzi D., Palmieri G., Costa D., Zingaretti S., Panebianco L., Ciuccoli N., Pinelli A., Callegari M., UNIVPM BRAVe: a hybrid propulsion underwater research vehicle. International Journal of Automation Technology, Vol. 11. No. 3, pp. 404-414, 2017
Page 3 - Figure 1: a vector or a simple arrow could be added to show the forward swimming direction
Page 3 - I understand that the prototype swims according to MPF locomotion exploting the undulation of its anal fins as Gymnotiform swimmers. However, readers which are not familiar with biologic propulsion could experience some difficulties while reading the paper. Therefore, I suggest to add these informations to improve the clarity of the paper.
Page 4 - Figure 2 (e) and (f) are a bit blurred
Page 5: As shown in Figure 3(b), the gravity of the...
I suggest to change "gravity" with "weight force"
Page 6: equations (7) and (8)
The values of these coefficients are reported in the following pages. However, i suggest to add a short formula to show how they are computed, or at least a proper reference.
Page 7:
Ignore the additional mass force
Assume that Coriolis force and centrifugal force have negligible influence on motion
A reference should be added to justify these assumptions
Page 11 - About the CFD analysis, I suggest to expand this section by adding further informations on the simulations, such as the number and type of elements and so on.
Page 12 - Figure 7: the picture is a bit blurred and should be improved.
Page 16 - Figure 11
I suggest to replace the label "Depth" with "Displacement"
Moreover, Figure 11 (b) is tough to undestand, particularly for the surge and sway displacements. As a matter of fact, the surge displacement seems to oscillate about the 0.6 value in the experiment chart, whereas it is rising - as expected - in the simulation chart.
I suggest to:
- move the attitude angle curve in a separate chart, like in figure (a).
- add the time-axis as in figure (a)
- display the correct trend of the surge displacement
Page 16 - Figure 12: is the velocity trend expected? Is there any correlation with the fin strouhal number?
Reviewer 2 Report
In general, this is a well written an informative paper. However, I think it is missing the scientific analysis that will make it a great paper.
The aim of the paper is not clear – what do the researchers wish to answer with this research? There is little to no discussion of what the results actually MEAN and so the conclusions seem a bit facile. I would strongly suggest taking out some of the extraneous detail on the control system (unless that is the novel contribution) and including a proper discussion section to address the following topics:
· How has the modelling approach improved or constrained the design and operation (results) of the craft? From a modelling point of view a cam is not flexible as the relative angle between sections is relatively permanent.
· What led to this decision and how what that angle chosen? Were the quadrotors a direct consequence of this choice (compare to real gymnotiformes) and how does that shape the results and performance that we can expect?
· How does this compare to other craft mentioned in the Introduction?
· What is the novel contribution? Is it the use of quardrotors as well as a ribbon fin (and if so, is that even a good idea?), is it the control system? If so, this needs to be explained? Is it the control strategy (and again if so, how is this affected by choices in the mechanism).
· Are the speeds reached by the craft “good” and if not, how do we judge if the craft is a success? Parameters for success need to be defined and quantified. Stability is mentioned but is not defined.
Content
The Abstract (and elsewhere), suggests that a single propulsion unit cannot achieve high efficiency. This feels untrue or perhaps disingenuous. Maybe this needs further justification or qualification but I would argue that by having more (inefficient) propulsion units, the efficiency is not improved.
The Introduction could and should be improved. Whilst the literature is relevant, the narrative around the literature is tedious. After reading it, I know that lots of work has been done but not why, nor what we (collectively) have learned or how it affects other research. This can be achieved by rewriting the introduction in a more thematic way and including more analysis (rather than just a list of achievements). A typical example of this is line 58: I know that Zhou has compared simulations and experiments but not why or how or even if it was considered successful. After such a sentence, you should explain to the reader why this is an important piece of literature to include.
At the end of the Introduction, I should be able to identify the key developments in the sector, how they relate to what has come before and know why they are all important.
Much research has been done on knifefish ribbon fins and there are more recent examples by Malcom McIver (among others) that address gymnotiform swimming, which may be more relevant. It is odd that the ribbon fin is taken as inspiration but supplemented by the four conventional quadrotors. Knifefish do not use these and can move in the vertical axis by running waves along the fin in opposing directions – this is not possible in your work owing to the cam arrangement. To me it seems strange to take inspiration from fish for one part of the design but constrain it in such a way that you need extra propulsion units. This decision making is what I would like to see discussed later in the paper.
Figures
The order of the figures does not always help to explain the points being made: If 2(c) is mentioned first, perhaps that should appear first. Figure 2(b) would be better with labels. Ray “chunk” is not the best word for this component (2(d)) and the labels in Figure 2(e) and 2(f) are too small. It may be better to separate these figures or to consider if they are really necessary.
Labels on Figure 3 are too small and the caption of 3(b) does not seem to describe the image.
Figure 6 (a) is too small to be useful. The first image should be labelled (a) if you keep these. The degree of the polynomial should be mentioned in the legend for Figure 6(b) and (c).
Figure 8 onwards: Graphs are probably too small to read clearly and the colours do not allow discrimination of all the series. Consider changing line styles and weights to improve clarity.
Figures 8, 9, 11 are hard to read as someone unfamiliar with the work. It is not always obvious which series should be compared. If we should be comparing the simulation with the experiment, it may be better to plot both those on one figure.
Figure 10 would be better with some labels or axes or even a stationary mark in each to show the progress/ Figure 11’s arrows work well in this respect.
Figure 12: Do you expect this relationship to continue or might we see a parabolic relationship between fin frequency and velocity (almost certainly!) – what does this MEAN? This also should be discussed properly.
Specific comments
Throughout:
There is a tendency to start sentences of definition with the verb not the subject, eg in line 208 “Define the control force matrix”. This should include the subject: “We define…”. Without the subject, it reads as an instruction rather than an account of what is being done. Please check for this throughout.
Numbers and units should be separated by a non-breaking space (CTRL+SHIFT+SPACE in Word, ~ in LaTeX), except for the degree symbol and the % symbol, which require not space. Please fix.
Line 12: by “unity” do you mean “unit?
17: What do you mean by “posture”?
43: What do you mean by “twisted”?
References to carangiform/BCF etc should include a picture or some way to explain this to the readers (I know what it means but I imagine the typical JMSE reader does not).
References to Section 2 etc in lines 88 –97 – Section should be capitalised.
91: space needed between Section and 2
120: How have you defined angle of attack in this instance? Does it relate to the local fluid velocity or to the free stream velocity?
154: What do you mean by pose?
157: You use u for one of the linear velocity components and later you use u (line 220) for the input force matrix. Please consider changing one of these.
161: Characters appear to be missing in the brackets of c(), s() and t() in the first part of the sentence.
166: Please check that Oi is defined somewhere.
174: Can you explain what you mean by symbolic function?
181: Can you define β on a diagram somewhere?
184: What do you mean by “approximately”?
189: Location of centre of buoyancy will not ensure it floats, please rewrite for clarity.
191: What does resilience represent in your model? Is it a physical quantity? Please explain.
254: What is a “functional”? Do you mean “function”?
259: Equation 18 has symbol missing between matrix and column vector (also a problem for equation 21)
261: Suggest “We know” rather than “We have known”
262: Is it quadratic or linear (surely it can’t be both)?
275: Table 1 could be moved to the appendix or at least moved so that it appears fully on one page not across two pages. It is not clear what the fin parameters relate to or how they were defined/measured.
277: “wind and other reasons” – was the tank outdoors? Another thing to discuss later in the paper.
279: Figure should be labelled as 5 not 1 and the text of (c) is too small to read – suggest you delete it.
285: “then” should not be capitalised.
305: Another Table 1! – please fix the numbering.
307 – 314: This could also be put in the appendix – it does not seem to add to the narrative.
316: schematic does not need a capital letter
327: Full stop (period) required at the end of sentence.
345 (351): It is preferable to work in millimetres and metres rather than centimetres.
351: Please insert a figure reference to the attitude angle curve.
361: How is stability assessed or defined?
369: What are the units of zero?
370-371: How do the propellers adjust the yaw angle? Are they angled to the Z axis or does some part of the robot act as a rudder?
391: Does the fin act as a rudder (or keel)?
394: Is the forward command made to the fin or the propellers? It is not clear in Figure 10 if the tilted ribbon fin is undulating or if the propellers are providing the force.
421: two commas in a row
425: This seems obvious
427: Stability remains undefined.
441: Why is such a short distance used?
446: Is a figure of 1.5 BL/s “good”? Put these measures in context by properly discussing your work with respect to your aims and to the literature!
470: Do you mean “increase” rather than “improve”?
Round 2
Reviewer 2 Report
Review v 2
I thank the authors for their thoughtful changes. The Introduction and the literature is much improved, although some proof-reading of the new sections is needed. I would advise the authors to put in a reference to a source describing the fish and motion types mentioned (for example there is good diagram in Colgate and Lynch, 2004, J Oceanic Engineering Vol 29, (3) which is very clear and as a review paper, it may also be interesting to the authors).
The manuscript would still be improved with more discussion on the major points given in the first review - there is rationale now in the introduction as to why these decisions have been made, which is very welcome, but the discussion should reflect on these decisions, take a broader view and ask "in hindsight, was this the correct approach?".
Some proof reading remarks - others left to editors
49: bent should be bend
52: St needs defining
54: et.al. formatting (no full stop needed after et)
87: Italics for latin names, ie G niloticus If a ventral fin inspiring fish is needed, perhaps you could look at Apteronotus albifrons instead for inspiration? Lots of research by RW Blake on knifefish.
147: system (lower case s), fails (with an s)
483: the “we” problem again
